# A High-Precision Remote Sensing Identification Method for Land Desertification Based on ENVINet5

**DOI:** 10.3390/s23229173

**Published:** 2023-11-14

**Authors:** Jingyi Yang, Qinjun Wang, Dingkun Chang, Wentao Xu, Boqi Yuan

**Affiliations:** 1Key Laboratory of Digital Earth Science, Aerospace Information Research Institute, Chinese Academy of Sciences (CAS), Beijing 100094, China; yangjingyi211@mails.ucas.ac.cn (J.Y.); changdingkun20@mails.ucas.ac.cn (D.C.); xuwentao221@mails.ucas.ac.cn (W.X.); yuanboqi23@mails.ucas.ac.cn (B.Y.); 2International Research Center of Big Data for Sustainable Development Goals, Beijing 100094, China; 3University of Chinese Academy of Sciences, Beijing 100049, China; 4Kashi Aerospace Information Research Institute, Kashi 844199, China; 5Key Laboratory of the Earth Observation of Hainan Province, Hainan Aerospace Information Research Institute, Sanya 572029, China

**Keywords:** desertification, land classification, fine classification, deep learning, influencing factors

## Abstract

Land desertification is one of the serious ecological and environmental problems facing mankind today, which threatens the survival and development of human society. China is one of the countries with the most serious land desertification problems in the world. Therefore, it is of great theoretical value and practical significance to carry out accurate identification and monitoring of land desertification and its influencing factors in ecologically fragile areas of China. This is conducive to curbing land desertification and ensuring regional ecological security. Minqin County, Gansu Province, located in northwestern China, is one of the most serious areas of land desertification, which is also one of the four sandstorm sources in China. Based on ENVINet5, this paper constructs a high-precision land desertification identification method with an accuracy of 93.71%, which analyzes the trend and reasons of land desertification in this area, provides suggestions for disaster prevention in Minqin County. and provides a reference for other similar areas to make corresponding desertification control policies.

## 1. Introduction

Desertification is a phenomenon of land sanding in arid and semi-arid areas caused by man-made and natural factors, i.e., the process of land gradually turning into desert. Desertification is a serious environmental problem that causes land resources to become unavailable and effective land resources to decrease, resulting in the loss and waste of resources, which in turn affects all aspects of economic development and human settlements and has attracted extensive global attention [1,2]. At present, there are many research studies on combating and mitigating desertification, and timely and accurate monitoring of the current situation of desertification and its dynamic changes will help to strengthen the prevention and management of desertification [3].

China’s desertification situation is very severe [4]. According to the survey results of the 6th National Desertification and Desertification Monitoring Bulletin [5], up to 2019, the area of desertification land in China was 2,573,700 km^2^, accounting for 26.80% of the total land area, and the area of desertification land was 1,687,800 km^2^, a net decrease of 37,880 km^2^ and 33,352 km^2^, respectively, compared with 2014. In 2019, the average vegetation coverage of desertified land was 20.22%, increasing 1.90% from 2014. Desertification land with vegetation coverage of more than 40% showed a significant increment, with a cumulative increase of 7,914,500 hectares in five years. Soil wind erosion in eight deserts and four sandy lands has generally weakened. In 2019, the total wind erosion was 4.179 billion tons, a decrease of 2.795 billion tons or 40% compared with 2000.

The central and northwestern regions of Gansu Province are located in the arid and semi-arid areas in northwestern China, with an annual precipitation of less than 200 mm, leading to the desertification land being widely distributed in the region [6]. The downstream area of Shiyang River Basin in the eastern part of Hexi Corridor in Gansu Province is a typical desertification area with harsh environmental conditions and a very fragile ecosystem. Land desertification in the downstream Shiyang River seriously affects the life quality of local people, economic development and natural ecological environment. So, quantitative research is urgently needed on the development trend of local desertification land, which is also an important means to effectively prevent and improve desertification.

Traditional desertification research mainly uses a manual field mapping method, which is time consuming, laborious and inefficient in the implementation process. Since the 1980s, remote sensing technology has developed considerably, and its applications have become increasingly widespread. Satellite remote sensing is capable of providing frequent and persistent surface information on the ground surface and is characterized by macroscopic, dynamic and precise monitoring of environmental changes on the surface. With the development of computer technology, machine learning methods, such as support vector machines (SVMs) and decision trees, are widely used in the classification and identification of remote sensing images [7,8]. Yan et al. (2013) used a decision tree and unsupervised classification to process the Landsat data in the study area and analyzed the desertification process in Mu Us sandy land over the past 40 years [9]. Taking Minqin oasis desertification area as the research area, Wang et al. (2000) extracted desertification land types by a decision tree [10]. Li et al. (2013) interpreted TM and ETM+ images of four periods in the study area by using spectral mixture analysis and a decision tree method and analyzed grassland desertification [11]. Liu et al. (2017) used MOD13Q1 as the data source and classified the degree of desertification in the Silk Road Economic Belt from 2000 to 2014 by using a normalized vegetation index and decision tree classification [12]. Li (2008) used a BP artificial neural network, combined with principal component transformation, and trained the neural network with several components containing effective information as input [13]. Qiao et al. (2004) established an automatic extraction model of land desertification information with visible light, thermal infrared and a normalized vegetation index in TM data as inputs of the BP neural network [14].

Traditional machine learning is widely used in land classification, but it also has some shortcomings, such as low accuracy, subjective influence on classification results and its internal process is invisible. Although the traditional model still has its applicability, the appearance of deep learning models in recent years has greatly improved the technical levels of image processing.

Deep learning is a branch of machine learning. By constructing a deep neural network model, a large amount of data can be quickly and accurately classified [15]. Compared with the traditional shallow neural network, the deep learning model has strong feature learning ability [16]. There are few related studies about its application in environmental monitoring and the application scope is still small, so it still has great potential for development. Shi (2018) used different convolutional neural networks, AlexNet, LCNet-27 and LCNet-13, to conduct a preliminary study on land cover classification using medium and high-resolution images and to perform a comparative analysis with traditional classification methods, which achieved good results [17]. Tian et al. (2018) used a convolution neural network and an SVM classifier to extract green space in Kubuqi desert [18]. Based on the hyperspectral image from the Gaofen-5 satellite, Sun et al. (2019) applied the U-NET deep learning model to study the classification of land use types. The overall classification accuracy was 0.9357, and the Kappa coefficient was 0.92 [19]. Therefore, deep learning models have achieved excellent performance in image classification, which can help us accurately determine the degree of desertification of land.

## 2. Study Area and Data

### 2.1. Study Area

Minqin is a county under the jurisdiction of Wuwei City, Gansu Province, China. It is located at the edge of the Loess Plateau, in the lower reaches of Shiyang River Basin (Figure 1). To the northeast and northwest, it is connected with Inner Mongolia. Its geographical coordinates are in the ranges of 38°04′07″ N–39°27′38″ N and 101°49′38″–104°11′55″ E, with 206 km from east to west and 156 km from north to south and a total area of 15,800 km^2^.

Because it is surrounded by desert on three sides (Figure 2), the climate of Minqin County is characterized mainly as continental desert climate [20]. In this area, the average precipitation is 110 mm, while with long sunshine hours, high accumulated temperature and vigorous evaporation, its annual evaporation is 2644 mm, which is more than 24 times its annual precipitation. In the summer it is very hot and dry, while in winter and spring it is cold with windy weather and strong sandstorm movement. Due to natural factors, natural disasters occur frequently in this region, which has a great impact on human life.

### 2.2. Data Resource and Preprocessing

As shown in Table 1, Population and economic data are from the statistical annual report of the official website of the Gansu Bureau of Statistics (http://www.minqin.gov.cn/, accessed on 1 June 2023).

The climate data were downloaded from China Meteorological Science Data Sharing Service Network (http://data.cma.cn/, accessed on 1 June 2023).

The remote sensing data were GF-6/WFV images, downloaded from the China Satellite Resources Application Center (https://www.cresda.com, accessed on 13 June 2022), with the acquisition date of 13 June 2022.

The GF-6/WFV data were preprocessed by radiometric calibration, atmospheric correction, ortho-rectification and vector clipping to obtain the surface reflectance data. Then, the NDVI (normalized difference vegetation index) and WBI (World Built-up Index) in the study area were calculated to be masked out from the region to obtain the preprocessed data, as shown in Figure 3.

## 3. Methods and Technical Flowchart

### 3.1. Technical Flowchart

As shown in Figure 4, the technical flowchart mainly includes data collection and preparation, data preprocessing, building a deep learning model, model training, model evaluation and optimization, prediction and classification.

(1)Data collection and preparation. Collect image data sets containing different types of land, including images of desertification and non-desertification land. These images can be obtained by satellite remote sensing data, drone images or on-site shooting. Ensure that the data set contains sufficient sample numbers and diversity.(2)Data preprocessing. Besides image format conversion, procedures such as size adjustment, brightness and contrast adjustment, and data enhancement operations, such as rotation, flipping and cropping, can be carried out to expand the data set and increase the generalization ability of the model.(3)Building a deep learning model. Choose a deep learning model suitable for land classification.(4)Model training. Use the prepared image data set to train the deep learning model. In the training process, the image is input into the model and the model parameters are constantly adjusted by the back propagation algorithm to minimize the classification error. You can use common deep learning frameworks, such as TensorFlow, for model training.(5)Model evaluation and optimization. The trained model is evaluated by using an independent verification data set. Calculation of the classification accuracy, precision, recall, F1 value and other indicators can be used to evaluate the model performance. If the model does not perform well, it can be improved by adjusting the model structure, superparameters or adding more training data.(6)Prediction and classification. Use the trained deep learning model to classify and predict the new land image. Input the image into the model, and the model will output the corresponding classification results to determine whether the land is experiencing desertification or not.

It should be noted that the performance of the deep learning model depended greatly on the quality and diversity of the training data. Therefore, reasonable data collection and preparation work are very important for building an accurate desertification land classification model. In addition, the desertification situation in different regions may be different, and it is necessary to optimize and adjust the model for specific regions.

### 3.2. Methods

As a complex form of machine learning, deep learning can make the system automatically discover the feature from the data. Compared with other machine learning methods, it can continuously improve the prediction accuracy without external guidance or intervention and draw conclusions through multi-layer learning in the neural network. For the processing of remote sensing images, the deep learning model attempts to discover and utilize the spatial features, spectral features and statistical features in remote sensing images [16].

The ENVINet5 model [22] was developed based on a Tensor flow deep learning framework. The model architecture is improved based on the U-NET neural network, which is similar to U-NET architecture, and it is an architecture based on mask-based or encoder–decoder systems. Combined with the powerful remote sensing, image processing software ENVI, it is convenient for researchers to directly use the deep learning network to process remote sensing images. Therefore, this study uses the ENVINet5 deep learning model to identify and classify the desertification land in the satellite images of the study area.

As shown in Figure 5, ENVINet5 model architecture has 5 levels and 27 convolution layers, and each level represents a different pixel resolution in the model. The original image is first inputted into the model. Then, the slice is convolved by a 3 × 3 convolution layer after slicing, which increases the number of image features. Using a 2 × 2 pooling layer, the size of the image is reduced and most of the image features are retained. The right part of Figure 5 shows the down-sampling process, and the left part shows the up-sampling process. Many features generated in the down-sampling process are classified by feature learning, recognition and classification, so as to achieve ground object classification.

After the remote sensing image is loaded into the deep learning network, it will be convolved after slicing. Convolution operation is essential to extract features from images and generate feature images with different dimensions, and it is often composed of multiple convolution kernels with different sizes. The commonly used convolution kernel sizes are 3 × 3 and 5 × 5, in which the convolution kernel size of this deep learning network is 3 × 3. In addition, the convolution operation is inseparable from the activation function. Activation function can not only simplify the complexity of operation but also speed up training and reduce the burden of computer operation to solve the problem of gradient disappearance. Rule activation function is often used in convolution operation to solve the problem of over-fitting in training. When the convolution operation is finished, a large number of characteristic images will be generated. If convolution operation is carried out again, the amount of operation will be greatly increased. Therefore, it is necessary to reduce the amount of computation while retaining more feature information of the generated feature image. The appearance of the pool layer has solved this problem well. Commonly used pools are mean-pooling and max-pooling. In this study, max-pooling is used with the window size of 2 × 2, which can not only extract the main features but also reduce the computational complexity. After pooling, the size of the feature image is reduced.

As shown in Figure 6, in order to extract the target, it is necessary to create a Label Raster that can indicate the target and then use the label sample to train the model. The Label Raster can be built from the classification task, the region of interest (ROI) task or the feature count tool. After the model is trained by the Label Raster, the trained model can then find targets with similar characteristics in other images. The extraction result is a Class Activation Map/Raster (CAM) in which the Digital Number (DN) value represents the probability that the pixel belongs to a certain ground object.

The initial classification raster and CAM may not be completely accurate, depending on the quality of the input training samples. An optional step is to refine the Label Raster by creating ROIs with the highest pixel values and then editing the ROIs to eliminate false positives. The refined ROIs can be combined with the original ROIs to retrain a new model or refine the trained model.

Before model training, an initialized tensorflow model is obtained by defining the relevant series of parameters. After that, the training parameters are set, and the initialized model is trained using the Label Raster to find the model with the best classification results. Finally, the trained model is used to locate the same features of the target image and classify the image.

## 4. Results

### 4.1. Desertification Identification Results

The results of desertification identification in this study area are shown in Figure 7.

From 5 March to 9 March 2023, we went to Minqin for field investigation and obtained 143 verified sample data (Figure 8).

After verifying the accuracy of the confusion matrix, the indicators included producer accuracy, user accuracy, total accuracy and the Kappa coefficient. As shown in Table 2, the accuracy of desertification land identification reaches 93.71%.

### 4.2. Dynamic Changes in Desertification

The Landsat time-series remote sensing satellite images from 2005, 2010 and 2015 were obtained, the vegetation coverage of Minqin County in each year was calculated (Figure 9). The desertification identification areas (Figure 10) and desertification degree (Table 3) were divided and analyzed dynamically.

The desertification area of Minqin County over the years was counted. As shown in Table 4, on the whole, the desertification area in the study area is the largest, and the degree of land desertification is mainly moderate. In 2015, the total area of land desertification in Minqin County was 15,000 km^2^. Compared with the survey area data of Minqin County in 2015, the two data are basically consistent, which shows that it is feasible to extract desertification information in Minqin County by using the statistical method of desertification area change.

In 2005, the area of heavily desertified areas reached a maximum of 2545.75 km^2^, while in 2010, it reached a minimum of 1764.38 km^2^, and in 2015, it was 2281.69 km^2^. It can be seen that the area of heavily desertified areas decreased in the past 15 years, but it was decreasing year by year before 2010 and rebounded in 2015.

The area of moderate desertification reached the maximum of 11,907.44 km^2^ in 2010, the minimum of 11,013.56 km^2^ in 2005 and 11,331.63 km^2^ in 2015. It can be seen that the area of moderate desertification has increased in the past 15 years, but the increment is not large.

The area of lightly desertified areas did not change greatly between 2005 and 2010 but decreased in 2010–2015. In 2010, the minimum area of non-desertification was observed at 358.31 km^2^, and the maximum area of 610.56 km^2^ was observed in 2015. On the whole, the area of non-desertification increased from 2005 to 2015.

The middle part of Minqin Oasis is dominated by mild desertification and non-desertification, while the periphery of Hong Yashan Reservoir is dominated by severe desertification and moderate desertification. In 2005–2010, the degree of severe desertification was reduced, and the area of moderate desertification increased, which may be related to the local implementation of measures such as changing farmland to forests and grasslands. From 2010 to 2015, the area of severe desertification increased, but from 2005 to 2015, there was a trend of gradual alleviation. Generally, land desertification in this region has been reversed in 10 years, but there is also a danger of expansion.

The Badain Jaran Desert and Tengger Desert in the county are mainly characterized by severe desertification and moderate desertification. Among them, the land desertification in the western region of the Badain Jaran Desert in the northwest of the study area was the most serious in 2005, which was the most serious scope of desertification in the study area. From 2005 to 2015, the desertification reversal phenomenon in this area was obvious and the degree of land desertification decreased. Other regions except the northwest region are dominated by moderate desertification, which accounts for the largest proportion in the research area of Minqin County and is also the main area of land desertification controlled in the research area. Generally, the distribution of different degrees of desertification in the whole study area is roughly similar in each year. Among them, the distribution of severe desertification is scattered, mainly in the northwestern Minqin County, south and north of the Hong Yashan Reservoir, and the intersection between Badain Jaran Desert and Tengger Desert. Moderate deserts are mainly distributed around the oasis, namely the Tengger Desert and Badain Jaran Desert in the county. Mild desertification and non-desertification areas are mainly distributed in oasis and oasis–desert transition zones.

### 4.3. Analysis of Influencing Factors of Desertification

The factors affecting land desertification can be divided into natural factors and human factors, and the development or reversal of land desertification is the result of the interaction between the factors [23]. The main natural factors in Minqin County are climate factors, such as precipitation and temperature, etc. Therefore, based on the meteorological, economic and population data of Minqin County, the influencing factors of land desertification in Minqin County are comprehensively analyzed from climate factors and human factors, respectively.

#### 4.3.1. Climate Factors

According to the climate data of Minqin County from 2005 to 2015, the annual average precipitation, annual average wind speed, annual average temperature and annual sunshine hours were selected as factors.

(1)Precipitation (Figure 11). The interannual variation is large, the precipitation is mainly concentrated in summer and the rain period is short. This aggravated the local drought, the lack of water in soil, the difficulty in the growth of surface organisms and the gradual decrease in the coverage areas in the region, which contributed to the conditions for the expansion of land desertification in Minqin County.

(2)Temperature (Figure 12). Rising temperature leads to increased evaporation, soil moisture loss and dry land. At the same time, the higher the temperature, the stronger the transpiration. In the dry season, the demand for water by plants is increasing, which leads to the continuous reduction in soil moisture, more obvious land desertification and further development of desertification.

(3)Wind speed (Figure 13). The wind speed determines the degree of wind erosion on the surface and its vegetation. Strong winds can erode the soil and gradually lead to land desertification. Strong winds blow up sand grains on the surface to form dust, which affects the growth of plants on the surface and damages agricultural production. Lower wind speed can reduce the pressure of vegetation restoration and desertification reversal.

(4)Sunshine hours (Figure 14). Sunshine hours affect the local accumulated temperature. The higher the accumulated temperature is, the stronger evaporation, which further leads to the reduction of water in the soil and the more serious desertification becomes. On the contrary, when evaporation reduces, the soil moisture is sufficient and the soil is sticky, which is not easy to erode and slows down the rate of desertification.

#### 4.3.2. Human Factors

According to the climate data of Minqin County from 2005 to 2015, the annual average precipitation, annual average wind speed, annual average temperature and annual sunshine hours were selected as factors.

(1)Economy (Figure 15)

The output value of various economic types in Minqin County is increasing year by year. After 2009, the growth of the primary industry is the slowest, but the output value has always been the largest among the three major industries. After 2009, the output value of the secondary industry has increased and the industry has developed rapidly, which will inevitably have a certain impact on the ecological environment of Minqin County. Over-exploitation and utilization result in scarce resources and abnormal ecological functions, and the desertification expansion in Minqin County plays a catalytic role.

(2)Population and grain yield (Figure 16)

The grain yield in Minqin County changes with the change in population, and the demand for grain is gradually decreasing with the continuous decrease in population. In 2009, the area of various types of desertification land in Minqin County reached 22.8 million acres. Due to water scarcity, 500,000 acres of artificial and natural vegetation died, and the harsh environment forced tens of thousands of Minqin people to relocate [24,25,26,27]. Because most of the permanent residents in Minqin County are middle-aged and elderly people, the decrease in grain output may force farmers and herdsmen to reclaim other grasslands to develop planting, and thus the abandoned farmland will be degraded under the influence of weathering. Furthermore, the unreasonable irrigation in the early stage will lead to secondary salinization of the soil. Lastly, due to the lack of understanding of environmental protection and the ecosystem, the fragile ecological environment is seriously damaged in the process of production and life, leading to serious land desertification.

(3)Animal husbandry (Figure 17)

From 2005 to the end of 2015, the production of sheep wool in Minqin County increased year by year, which indicates that the number of sheep kept by the people of Minqin County has increased year by year.

With the development of animal husbandry, if the livestock carrying capacity of grasslands is neglected, the expansion of sheep flocks will probably lead to severe grazing of surface vegetation, delayed recovery of pastures, declining pasture yields, and serious degradation of grasslands, which will promote the expansion of land desertification.

(4)Policy

Since 1959, Minqin Oasis has begun to plant artificial oasis shelterbelts such as Elaeagnus angustifolia and Haloxylon ammodendron on a large scale to prevent land desertification and restore the ecosystem [28]. However, after the 1980s, due to the extensive land reclamation, the artificial oasis shelter forest experienced a large-scale decline and death. Since 2007, Minqin County has implemented the project of returning farmland and grazing land to forests and grasslands [29]. The Key Management Plan of Shiyang River Basin [30] was also implemented in 2007. The Minqin water transfer project was implemented, whose distribution scheme of Shiyang River was determined, and the water storage in Minqin County was increased to meet the overall water demand of Minqin County, and thus reducing the area of desertification land to some extent.

#### 4.3.3. Quantifiable Indicators of Climate and Human Factors

The quantifiable indicators of climate and human factors in Minqin County from 2005 to 2015 were selected by principal component analysis. The climate factors include the annual average wind speed (x1), the annual average temperature (x2), the annual precipitation (x3) and the annual sunshine hours (x4). The human factors include the annual grain yield (x5), the annual wool production (x6), the annual population (x7) and the annual GDP (x8).

Table 5 gives the eigenvalue, contribution rate and cumulative contribution rate of each component. According to the conditions that the eigenvalue is greater than 1 and the cumulative contribution rate reaches about 80%, two components are selected as the main factors from 2005 to 2015. The weight of each factor is obtained from the selected principal factor, and the factor rotation operation is carried out. From this, Table 6 is obtained and the following conclusions are drawn: during the period of 2005–2015, the annual GDP, the annual wool production, the annual population and the annual grain yield have a large weight on the first principal factor, accounting for a large proportion, while the annual precipitation, annual sunshine hours, annual average wind speed and annual average temperature have a large weight on the second principal factor.

In summary, land desertification in Minqin County is the result of the combined action of climate factors and human factors, but human factors are the main influencing factors. In Minqin County, climate factors are relatively stable, but human factors have subjective characteristics. In fact, the effect of human factors on land desertification is completed with the help of specific climatic conditions. When human factors play a role in promoting the development of desertification, it naturally destroys the local natural environment, and conversely, the harsh natural environment will also affect the local social and economic development. Only by the influence of climatic conditions, will the process of land desertification slowly change. Under the influence of better climate factors, desertification areas can increase in water content with the help of appropriate temperature and precipitation, thereby maintaining strong soil fixity and the growth of surface vegetation and thus gradually restoring its ecological functions. However, the combination of unreasonable human activities and deteriorating climatic conditions increases the fragility of the ecological environment and accelerates the development of desertification. In addition, people are forced to increase livestock slaughter, putting pressure on the carrying capacity of the grassland and leading to the unreasonable use of other land resources, which will also not only change the ecological environment but will also affect the growth of surface vegetation and indirectly promote the formation of desertification land. For the land that has been subjected to desertification, it is necessary to restore its ecological function with the help of good climatic conditions. However, due to the intervention of human activities, the degree of degradation has been strengthened, leaving a lot of land exposed, destroying the grassland and making the degree of land desertification more serious.

In recent years, the state and local governments have carried out a series of ecological protection and restoration projects to effectively improve vegetation coverage and reverse desertification land, which has made great achievements in local economic development. From the factor analysis, it can be seen that the development of economic and animal husbandry account for a relatively large weight, which is the thrust that affects the reversal of desertification. We should make rational use of land resources to ensure the economic income of farmers and herdsmen and take measures to increase land shortage.

Generally speaking, for areas with severe desertification, measures such as ecological migration, land closure and grass cultivation should be taken. For areas with light desertification, measures such as rotational grazing should be implemented to maintain the ecological balance of grassland, land use patterns, and develop planting and aquaculture in multiple ways while ensuring that the scope of desertification land is not expanded, which not only solves the economic income problem of farmers and herdsmen but also fundamentally solves a series of blind and unreasonable economic behaviors.

## 5. Conclusions and Prospect

In this paper, Minqin County was selected as the experimental area of this study. The high-scoring satellite images were obtained, and the land was classified by using an EN-VINet5 model combined with the field investigation and the sampling point data, and the desertification area was extracted based on the classification results. After verification, the accuracy was 93.71%. Compared with the traditional machine learning classification method, the deep learning method used in this study can automatically extract the features of remote sensing images, so it can effectively avoid complicated work such as manual extraction and feature selection and has high operability.

By analyzing the influencing factors, the driving mechanism of desertification can be better discovered. Due to the comprehensive influence of climate factors and human activities, desertification has become an important environmental problem in Minqin County which is located in the Loess Plateau, with arid climate, scarce precipitation and easy land degradation. At the same time, unreasonable human activities also have an irreversible impact on the land environment. For example, unsustainable farming and animal husbandry practices, such as overgrazing, deforestation and over-cultivation, have aggravated the land degradation.

Human factors are the dominant influence on land desertification in Minqin County. In order to deal with the problem of desertification, Minqin County has taken a series of measures. First of all, the county is strengthening land management and protection to control overgrazing and overexploitation; secondly, it is promoting vegetation restoration and protection by taking measures such as afforestation and grassland restoration to increase vegetation coverage and to improve soil water and fertilizer conservation capacity. At the same time, it is strengthening the construction of soil and water conservation projects to prevent soil erosion and loss. In addition, the government also actively guides farmers to change agricultural management methods to promote water-saving irrigation and sustainable agricultural modes and to reduce excessive pressure on the land.

Although Minqin County has made some achievements in desertification control, it still faces some challenges. Desertification is a long-term and complicated process, which requires the joint efforts and continuous input of the whole society:(1)Intensive land utilization can effectively reduce the negative impacts of land desertification. On one hand, land control actions have been vigorously carried out to restore surface vegetation, consolidate soil and fix soil so that the ecosystem can develop in a balanced manner and in a virtuous cycle. On the other hand, the government strengthens the scientific division of functional areas of land use to promote the harmonious development of agriculture, forestry and animal husbandry, which is conducive to the sustainable development of land;(2)Optimizing the economic and industrial structure to avoid damage to the natural environment caused by the excessive pursuit of industrial and economic growth, and thus to promote the harmonious symbiosis of economic development and environmental protection;(3)Strengthening investment in scientific research related to desertification control technology, utilizing modern science and technology to slow down or even stop the trend of land desertification, and further improving the effectiveness of land desertification control.

The government, scientific research institutions, farmers and other participants should not only strengthen cooperation but also intensify scientific and technological innovation and formulate more scientific and feasible desertification control strategies to realize sustainable land management and protection, which would promote the recovery of the ecological environment and the sustainable development of Minqin County.

## Figures and Tables

**Figure 1 sensors-23-09173-f001:**
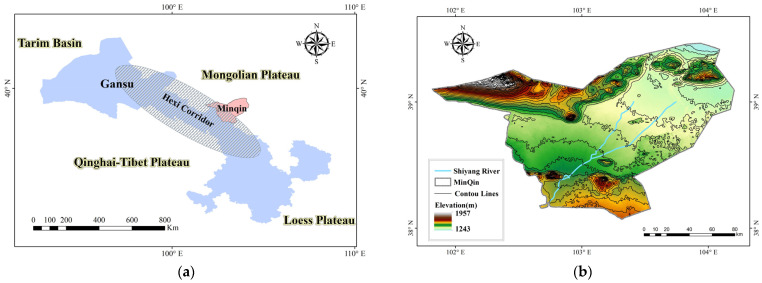
Location of Minqin County with DEM: (**a**) Location map; (**b**) Digital Elevation Map (DEM).

**Figure 2 sensors-23-09173-f002:**
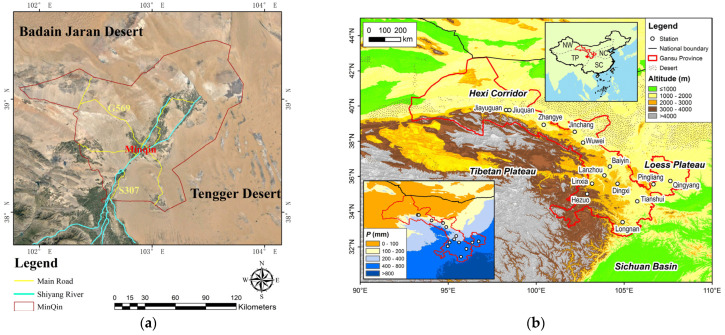
Spatial distribution of Minqin County and location of Gansu Province [21]: (**a**) Minqin County; (**b**) Gansu Province.

**Figure 3 sensors-23-09173-f003:**
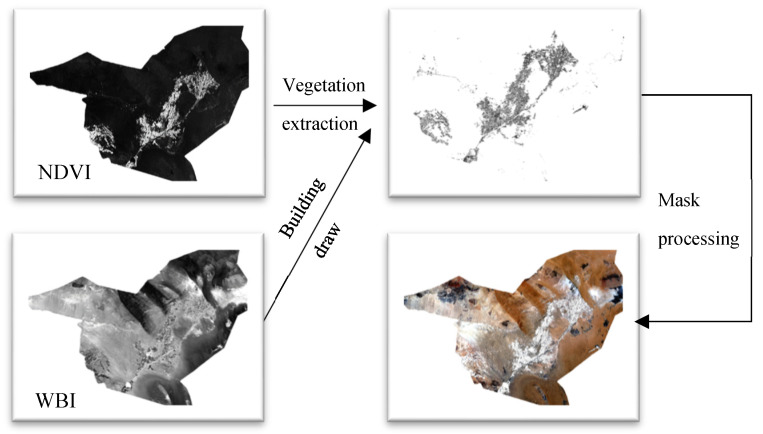
Mask preprocessing of remote sensing data.

**Figure 4 sensors-23-09173-f004:**
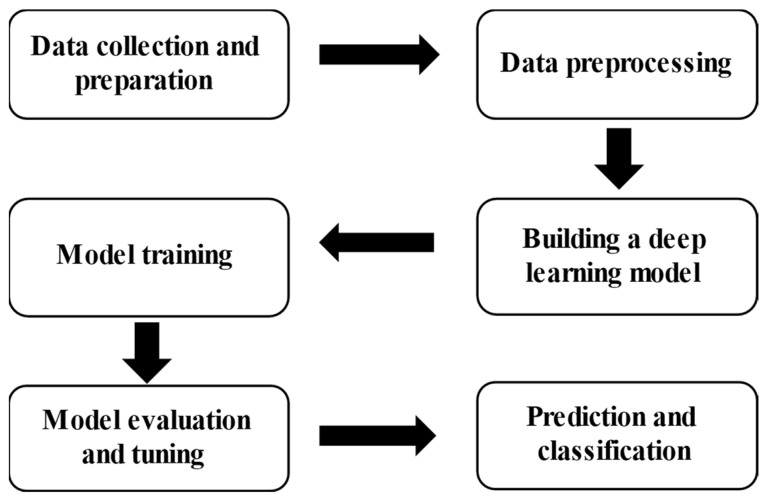
Technical flowchart.

**Figure 5 sensors-23-09173-f005:**
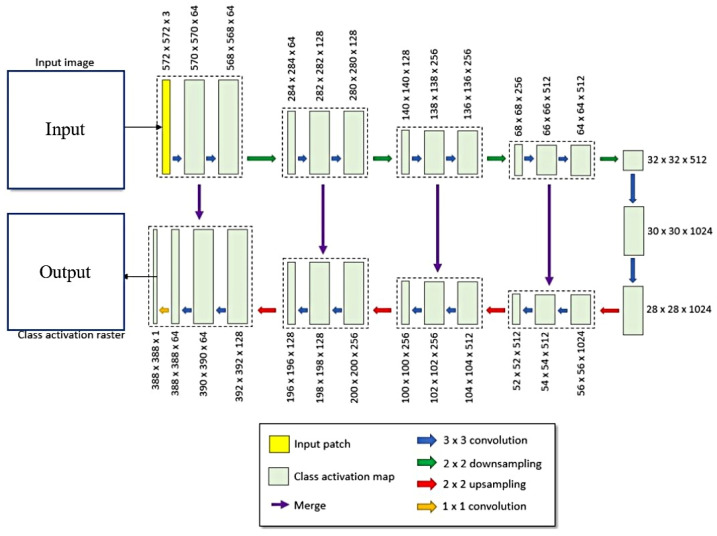
ENVINet5 model architecture diagram.

**Figure 6 sensors-23-09173-f006:**
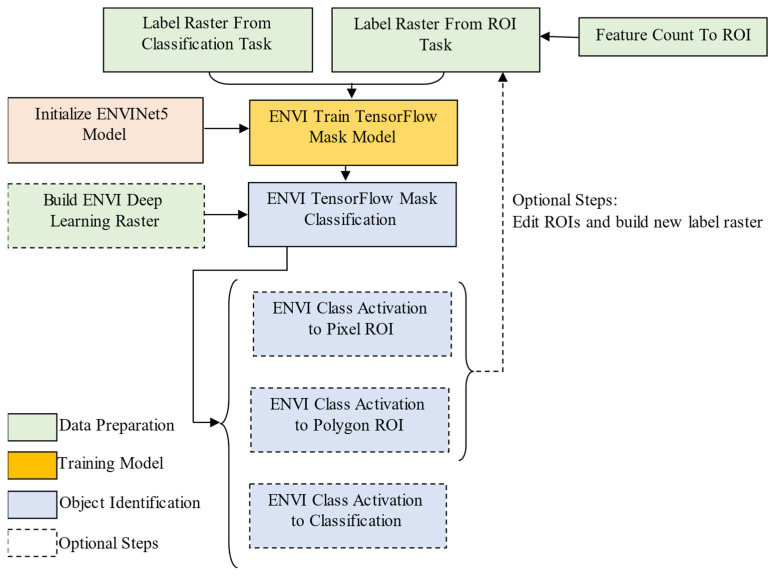
Flow chart of deep learning classification.

**Figure 7 sensors-23-09173-f007:**
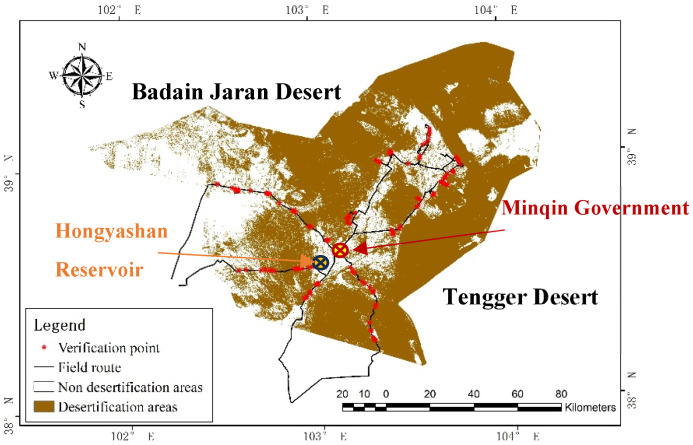
Map of desertification land in Minqin and field investigation route.

**Figure 8 sensors-23-09173-f008:**
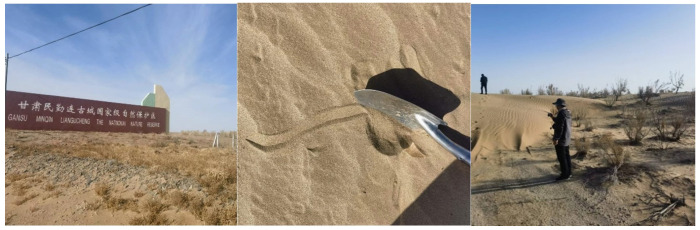
Field investigation in Minqin County.

**Figure 9 sensors-23-09173-f009:**
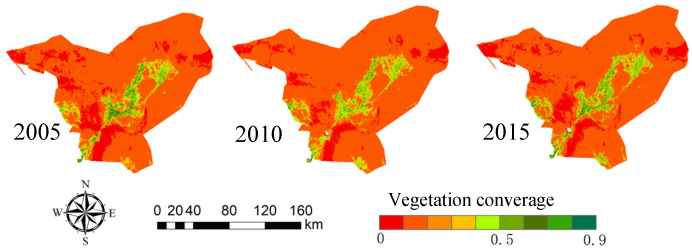
Vegetation coverage in Minqin County from 2005 to 2015.

**Figure 10 sensors-23-09173-f010:**
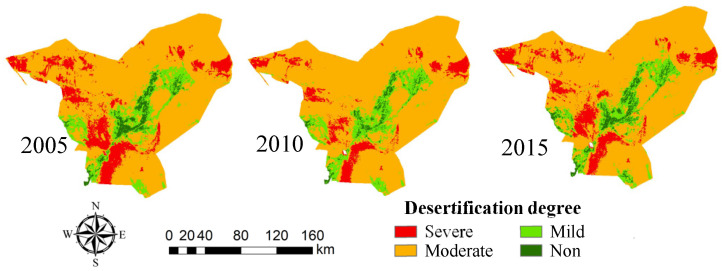
Degree of Land Desertification in Minqin County from 2005 to 2015.

**Figure 11 sensors-23-09173-f011:**
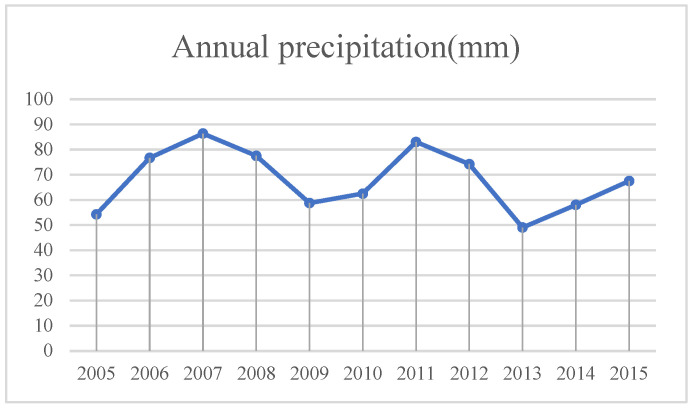
Annual precipitation in Minqin County from 2005 to 2015.

**Figure 12 sensors-23-09173-f012:**
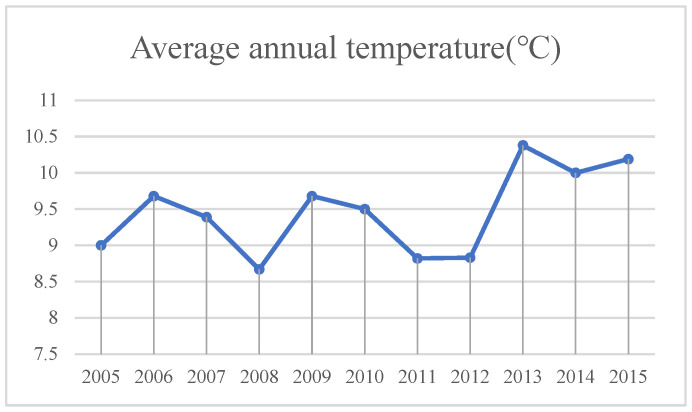
Average annual temperature in Minqin County from 2005 to 2015.

**Figure 13 sensors-23-09173-f013:**
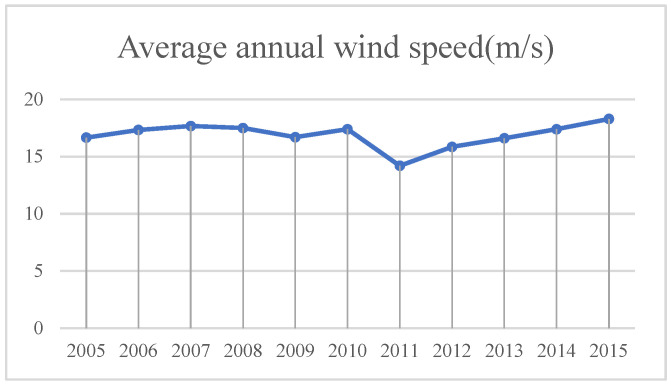
Average annual wind speed in Minqin County from 2005 to 2015.

**Figure 14 sensors-23-09173-f014:**
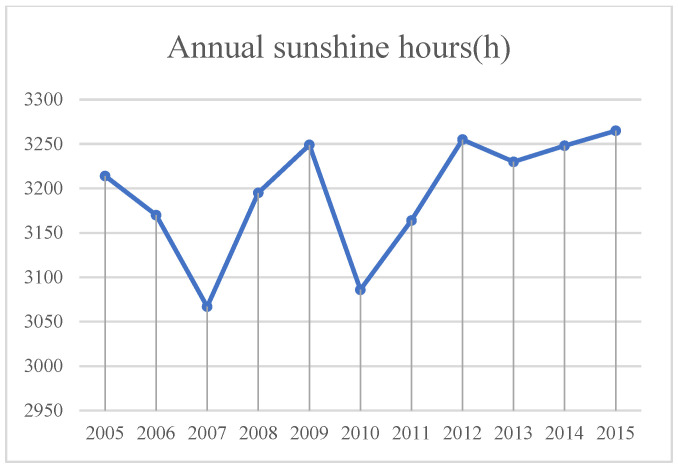
Annual sunshine hours in Minqin County from 2005 to 2015.

**Figure 15 sensors-23-09173-f015:**
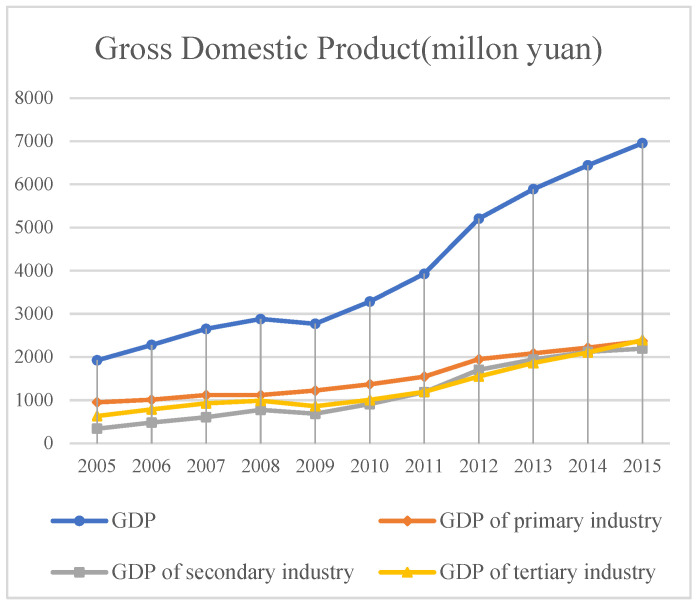
Gross Domestic Product of Minqin County from 2005 to 2015.

**Figure 16 sensors-23-09173-f016:**
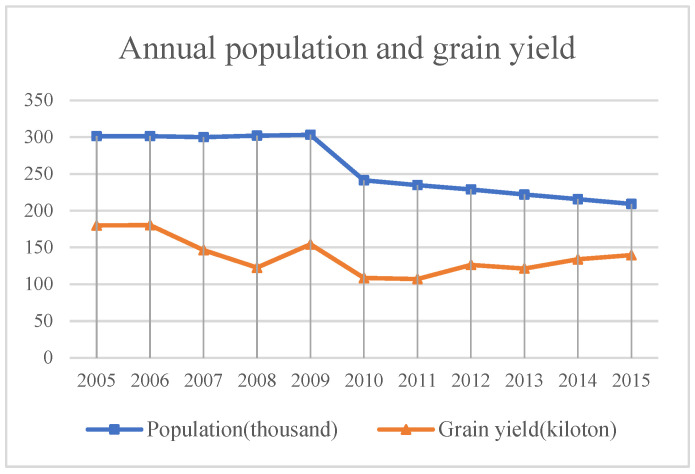
Cultivated land area in Minqin County from 2005 to 2015.

**Figure 17 sensors-23-09173-f017:**
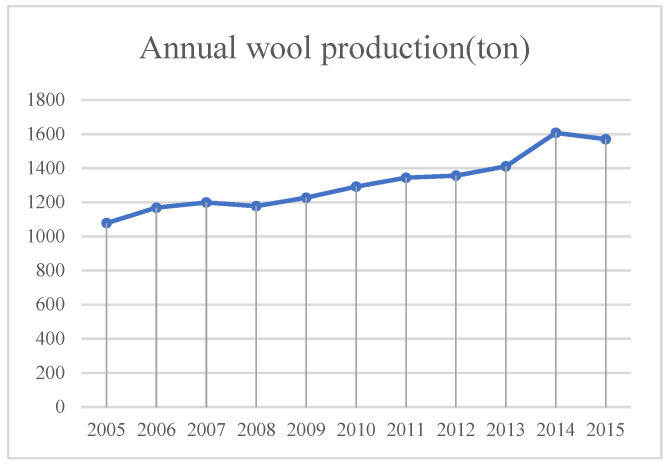
Annual wool production in Minqin County from 2005 to 2015.

**Table 1 sensors-23-09173-t001:** Data resource.

Data	Downloaded Date	Resource	Resolution
Demographic and economic data	1 June 2023	Gansu Statistics Department	
Climate data	1 June 2023	http://data.cma.cn/	
DEM	1 June 2023	SRTM	30 m
GF-6/WFV	13 June 2022	https://www.cresda.com	16 m

**Table 2 sensors-23-09173-t002:** Accuracy Verification of Confusion Matrix.

Recognition Result	Verification Data			
	Non-Desertification	Desertification	Amount	User Accuracy
Non-desertification	59	0	59	100%
Desertification	9	75	84	89.29%
Amount	68	75	143	
Producer accuracy	86.76%	100%		
Overall accuracy				93.71%
Kappa				0.87

**Table 3 sensors-23-09173-t003:** Classification of desertification degree.

Vegetation Coverage	Desertification Degree
<0.1	Severe
0.1–0.2	Moderate
0.2–0.5	Mild
>0.5	Non

**Table 4 sensors-23-09173-t004:** Statistics of the Desertification Area in Minqin County from 2005 to 2015 (km^2^).

Time	Severe Desertification	Moderate Desertification	Mild Desertification	Non-Desertification	Total Area
2005	2545.75	11,013.56	1729.63	537.06	
2010	1764.38	11,907.44	1795.94	358.31	15,826
2015	2281.69	11,331.63	1599.13	613.56	

**Table 5 sensors-23-09173-t005:** Eigenvalue and factor contribution rate table.

Component	Initial Eigenvalues	% of Variance	Cumulative %
1	4.126	39.48	39.48
2	2.771	26.52	66.00
3	1.534	14.68	80.68
4	0.926	8.86	89.54
5	0.598	5.51	95.05
6	0.297	2.84	97.89
7	0.135	1.29	99.19
8	0.063	0.60	100

**Table 6 sensors-23-09173-t006:** Factor weight table after rotation.

Variable	Component	
	1	2
Annual average wind speed	−0.164	0.704
Annual average temperature	0.583	0.791
Annual precipitation	0.201	−0.787
Annual sunshine hours	0.064	0.802
Annual grain yield	−0.751	0.137
Annual wool production	0.927	−0.064
Annual population	−0.859	0.009
Annual GDP	0.982	0.201

## Data Availability

Publicly available datasets were analyzed in this study. The data sources can be found in Section 2.2.

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
