# Peer review of "A High-Precision Remote Sensing Identification Method for Land Desertification Based on ENVINet5"

_sensors, 2023, doi:10.3390/s23229173_

Round 1

Reviewer 1 Report

Comments and Suggestions for Authors

Authors have made an attempt to classify the land cover classes with the help of ENVINet which was introduced recently in the ENVI Deep learning module. However, some of the major observations are as follows:

1.      In the Title, the “research” word is unsuitable as it is obviously a research paper. Moreover, it is better to define the type of deep learning model used in the present article.

2.      the type of deep learning model is also not mentioned in the abstract. If this article involved the comparison, then the authors must mention it in the abstract along with the results. So the significance of the deep learning model will be highlighted.

3.      Very little information is provided with respect to the deep learning model. I think the authors have explained them in more detail.

4.      L107, please recheck the degree sign in the coordinates.

5.      More information or technical information is required under the section “Data” such as technical details of datasets utilized in the manuscript.

6.      In Fig. 1, abbreviations must be specified in the footnotes of the figure. Moreover, the overlap of arrows on the extraction.

7.      The subsection must be refined by putting all the points into the form of paragraph w.r.t Fig. 3. Fig. could be refined with more details of the Deep learning model.

8.      Fig. 4 needs to be replaced with high resolution image.

9.      All the figures must be replaced with high resolution as these are not visible on the printed version.

10.   No need to put the references under the section “Conclusion”. It is suggested that add most of the details of the conclusion in the discussion part. 

Reviewer 2 Report

Comments and Suggestions for Authors

Dear Authors,

I find your manuscript an interesting and important contribution with novel approaches. The manuscript is generally well structured, the research design is fine but still tehre is some room for development. Probably not the main frame of the work that needs some revision, but the refinement of the details need work on. There are way too many sections there, a little extra input the report would be more convincing. One of the main issues is the totally underutilization of figure captions. Most of the captions are very random and short and do not really provide the needed information for the reader to understand the figures without deeply reading the main text. This must be revised carefully. Also, the figures are very low quality. Not only resolution issues there but basic cartography issues as well that is strange in a manuscript that is based on remote sensing. You also mentioned a lot of geographical location but there is no China overview map, or the detailed maps has not showed all the mentioned locations. Map figures also need to be referred to more frequently in the main text.

Overall major revision is inevitable. This also should include another run of English editing for clarity. I have provided a detailed annotated PDF to show where I can see some room for improvement.

Best regards,

Comments on the Quality of English Language

English is fine but a good final English editing of the revised manuscript would be good

Round 2

Reviewer 1 Report

Comments and Suggestions for Authors

The authors have made significant improvements in the revised manuscript.

Reviewer 2 Report

Comments and Suggestions for Authors

Dear Authors,

Thank you for your revised version submitted to Sensors (MDPI). I have checked the revised version independently and in the light of the previous comments over some issues. It seems the revised version addressed well all the issues and questions raised. The revised version also improved the presentation style and focused more on the explanation of the method. The revised version at the level of the standard Sensors requires for publication hence I can recommend your work to be accepted as it is. I have no further queries.

Best regards